# The Incidence of Severe Hypercalcaemia-Induced Mental Status Changes in Patients Treated with Antibiotic-Loaded Calcium Sulphate Depot for Orthopaedic Infections

**DOI:** 10.3390/jcm11164900

**Published:** 2022-08-20

**Authors:** Faustine Vallon, Christoph Meier, Emanuel Gautier, Peter Wahl

**Affiliations:** 1Division of Orthopeadics and Traumatology, Cantonal Hospital Winterthur, 8401 Winterthur, Switzerland; 2Department of Orthopaedics, HFR Fribourg-Cantonal Hospital, 1752 Fribourg, Switzerland

**Keywords:** calcium sulphate, CaSO_4_, local antibiotics, carrier material, hypercalcaemia

## Abstract

Local application of antibiotics with calcium-containing carrier materials (CCCM) might deliver large quantities of calcium, with some cases of hypercalcaemia reported. The incidence of symptomatic hypercalcaemia was estimated retrospectively in a consecutive, prospective series of patients treated between 10/2006 and 02/2019 with antibiotic-loaded CCCM for various orthopaedic infections. Risk factors were analysed. In the study period, 215 CCCM applications were performed. Two patients (0.9%) developed symptomatic hypercalcaemia. In one case, hypercalcaemia occurred 14 days after a second CCCM application during a staged septic hip revision. In the other case, hypercalcaemia became symptomatic six days after application of vancomycin-loaded CCCM in a component-retaining septic revision hip arthroplasty. In both cases, hypercalcemia was not imputable solely to the CCCM. Prolonged immobilization, renal impairment and other specific risk factors were present. Implantation of a CCCM for local application of antibiotics exposes the patient to large quantities of calcium during dissolution. This might induce symptomatic hypercalcaemia, a potentially life-threatening complication. The observed incidence of symptomatic hypercalcaemia remained rare (<1%). In some patients, compensatory mechanisms might be overwhelmed in the presence of other risk factors. Postoperative monitoring of calcaemia as well as elimination of risk factors is mandatory for all patients treated with CCCM.

## 1. Introduction

Calcium sulphate (CaSO_4_) is a bone graft substitute well known to be useable, either in pure form or as a composite, as a carrier material for local delivery of antibiotics. It has been applied to treat osteomyelitis or periprosthetic joint infection (PJI) [1,2,3,4,5,6]. It has also been used to enhance treatment of non-orthopaedic implant-associated infections [7]. Compared to polymethylmethacrylate (PMMA) bone cement, it offers some relevant advantages as antibiotic carrier material. As it dissolves in vivo, no secondary surgery is necessary for removal [2]. Vancomycin is released from CaSO_4_ for a longer period of time and with local concentrations ten times higher than from PMMA [5,8,9,10]. Vancomycin levels remain above staphylococcal biofilm eradication thresholds for up to two weeks and are up to three months post-operatively above common minimum inhibitory concentrations for Staphylococcus [5,11].

The dissolution of calcium-containing carrier materials (CCCM) may, however, expose the patients’ metabolism to a large calcium load [12,13]. The first concerns about hypercalcaemia induced by implantation of CaSO_4_ were published in 2005 as a United States Food and Drug Administration (FDA) adverse reaction report [14]. To the best of our knowledge, only three cases of symptomatic severe hypercalcemia have been published so far [15,16,17]. No data regarding incidence and risk factors of this potentially life-threatening complication are available. It was the aim of the current retrospective study to identify cases of symptomatic hypercalcaemia potentially related to application of CCCM for the treatment of bone and joint infections, retrospectively analysing a prospectively collected series with over 200 applications. Two cases of hypercalcaemia-related mental status alterations could be identified. As illustrated, the aetiology of hypercalcaemia may be multifactorial, and is not necessarily linkable directly to the implanted CCCM.

## 2. Patients and Methods

This is a retrospective analysis of a prospectively maintained patient registry. The registry enrolled all patients in whom CaSO_4_ (Osteoset, Wright Medical, Memphis, TN, USA) had been implanted between October 2006 and February 2019 as a carrier material for local delivery of antibiotics in the treatment of various orthopaedic infections. This patient register was prospectively collected to monitor a therapy newly introduced at that time for bone and orthopaedic implant-associated infections which were treated with antibiotic-loaded calcium sulphate in this two departments. All patients in the above-mentioned time period who received CCCM were included in the study. All patients provided written informed consent for anonymized analysis of the data. Routine postoperative monitoring was described elsewhere [5,18]. Monitoring of serum calcium levels had been performed only in case of clinical suspicion.

Identification of symptomatic hypercalcemia was achieved as available in the electronic patient files, but was not limited to the explicit diagnosis in the various reports. Complementary investigations and treatment of major alterations of consciousness, stuporous state or any other major alteration of consciousness were specifically checked for hypercalcaemia by searching for results of serum calcium assays, either blood gas analysis or from venipuncture. The same was completed for cardiac dysrhythmia. The renal glomerular filtration rate was estimated using the Cockcroft–Gault equation, modified considering height and BMI [19].

## 3. Results

Only 2 out of the 215 patients (0.9%) of this case series were identified as having developed symptomatic hypercalcaemia. Demographic and surgical data of the whole cohort as well as of the patients with hypercalcemia are listed in Table 1. Details of these two cases are described below. The evolution of the calcaemia over time is illustrated in Figure 1. Systemic antibiotics given to these two patients are illustrated in Figure 2.

### 3.1. Case One

A 68-year-old male active smoker developed septic arthritis with *S. aureus* after externally performed repeated glucocorticoid infiltrations to his left hip for symptomatic osteoarthritis. Resection arthroplasty of the hip was performed emergently, considering sepsis and advanced degenerative alterations of the joint, and intravenous antibiotic therapy was started after sampling.

The patient developed moderate renal failure (eGFR ~50 mL/min/1.7 m^2^) on the first postoperative day. During a second look procedure, the dead space of the hip joint was filled with CaSO_4_ beads loaded with vancomycin (75 mL Osteoset standard cure with 6 g vancomycin Labatec, Labatec Pharma, Meyrin, Switzerland). Sixteen days after admission, the clinical evolution was slowly improving, allowing planning of total hip arthroplasty (THA). However, the patient developed interstitial nephritis caused by penicillin and antibiotic treatment was modified (Figure 2). Additionally, the patient developed urinary sepsis with *E. coli* and had to be admitted to the intensive care unit (ICU) due to development of encephalopathy combined with acute renal failure (eGFR 39 mL/min/1.7 m^2^). Therefore, reconstruction was delayed.

Almost a month after admission, conventional radiographies showed partial resorption of the CaSO_4_ beads and the patient’s condition had improved for THA to be performed, without an antibiotic free interval. The local vancomycin level was 61.3 mg/L at that time, consistent with known kinetics [5]. Fifty millimetres of CaSO_4_ beads with a total of four grams of vancomycin were added to optimize pre-emptive treatment of any bacterial persistence. Five days after THA, the wound was dry, and nine days after THA, the patient was transferred to an inpatient rehabilitation facility.

Five days later, the patient had to be readmitted due to severe symptomatic hypercalcemia (albumin-corrected Ca^++^ 4.43 mmol/L) combined with altered mental status, vomiting, constipation and acute prerenal failure (eGFR 36 mL/min/1.7 m^2^). Retrospectively, hypercalcaemia had already been present in the blood gas analysis right after THA (ionized Ca^++^ 1.42–1.72 mmol/L) but had not been symptomatic at that time and was overlooked. Under forced hydration for several days and after a unique administration of intravenous bisphosphonates (4 mg zolendronate) and of 500 units calcitonin subcutaneously, calcemia normalized within 14 days (Figure 1).

Extensive investigations excluded an osteolytic disease, as well as a paraneoplastic or granulomatous aetiology. It, however, appeared that the patient had a habitual exaggerated calcium intake, due to consumption of 3 L of milk per day. Prolonged immobilization after the initial resection arthroplasty, followed by partial weight-bearing after THA combined with recurrent renal failure, represented a multifactorial aetiology in a patient’s metabolism already stressed by the calcium burden secondary to the dissolution of the CCCM implanted in the hip.

### 3.2. Case Two

An 82-year-old (152 cm, 54 kg) female patient had repeated revision THA due to loosening of the cup and later due to recurrent dislocation. Past medical history included resection of a parathyroid adenoma four years earlier due to primary hyperparathyroidism and a subvalvular aortic stenosis. Follow-up examinations were uneventful with normal serum calcium levels, without further medication. Osteoporosis was treated with vitamin D only. Approximately four weeks after her last hip surgery, the patient was emergently readmitted due to PJI. Joint aspiration identified *S. epidermidis*. Considering the short interval from operation to development of infection, the complexity of the previous acetabular reconstruction, the poor general condition and the cardiac risk profile, an implant-retaining treatment was favoured. This approach was supported by the fact that the causative staphylococci were rifampin-sensitive and, therefore, biofilm-active antibiotic treatment was available. Debridement with exchange of the modular components with the adjunct of CaSO_4_ beads (75 mL Osteoset standard cure loaded with 6 g vancomycin, Vancomycin Labatec) was performed. Systemic antibiotic therapy with intravenous daptomycin was initiated.

On the sixth postoperative day, the patient became confused. Hypercalcemia (albumin-corrected Ca^++^ 4.21 mmol/L) was detected. Calcaemia responded well to intravenous hydration, administration of intravenous bisphosphonates (4 mg of zolendronate) and subcutaneous calcitonin at 8 IU/kg of body weight (Figure 1). Despite this treatment, the neurologic status deteriorated, the patient becoming stuporous. Three days later, increased creatinine serum levels and an eosinophilia were observed. Daptomycin was immediately ceased and treatment switched (Figure 2). The altered mental status recovered rapidly. Serum calcium normalized within 12 days. Both daptomycin neurotoxicity and hypercalcaemia had to be considered etiologic for the neurologic alteration. The patient could be transferred to an inpatient rehabilitation facility seven weeks after admission.

## 4. Discussion

Hypercalcaemia may be a life-threatening condition, associated with polyuria, polydipsia, vomiting, hyperpyrexia, cognitive disorders, somnolence or coma, and usually associated with total calcium concentrations of >3.5 mmol/L [20]. It may be induced by tumours (e.g., paraneoplastic or secondary to osteolysis), drugs, endocrine disturbances, immobilization, granulomatous disease (e.g., sarcoidosis) or, rarely, familial hypocalciuric hypercalcaemia [20]. Implantation of CCCM for local antibiotic therapy in orthopaedic infections was also reported as being causative [15,16,17]. Despite the rather large amount of calcium dissolving from CCCM within weeks to months after implantation, hypercalcaemia-related mental status alteration was observed in only 2 out of 215 (0.9%) applications of antibiotic loaded CaSO_4_ in the treatment of various bone and joint infections (Table 1). In both patients, other risk factors contributing to hypercalcaemia were present. Once recognized, the serum calcium could be corrected successfully, with no recorded recurrence.

However, no systematic monitoring of serum calcium levels was performed in this case series. Serum calcium levels were measured only based on clinical suspicion. Thus, asymptomatic hypercalcaemia related to CCCM might be more frequent. No hypercalcaemia-associated dysrhythmias were identified. In addition, less specific symptoms such as polydipsia, polyuria or intestinal cramping are simply impossible to identify retrospectively in such a cohort of patients, requiring perioperative analgesia and antibiotic treatment. One could argue that the incidence of symptomatic hypercalcaemia might also have been higher, with all patients not necessarily identified, primarily with neurologic symptoms being missed or attributed to other etiologies. The course of CCCM-related hypercalcaemia is not known. Spontaneous recovery without treatment may occur. However, all our patients attended regular orthopaedic follow-up examinations in the same institutions for at least one year after discharge. Both hospitals serve as regional reference centres with ICU, where affected cases should have been admitted, if not recognized previously. It may thus be reasonable to assume all severe symptomatic cases should have been identified.

Both reported cases illustrate that many contributing factors have to be considered. Hypercalcaemia might occur late after implantation of CaSO_4_, when most of the material has already dissolved. In Case One, it was over 30 days after the initial operation and 14 days after the second implantation of CCCM. In Case Two, the patient developed symptomatic severe hypercalcaemia 6 days postoperatively. Published reports describe the occurrence of severe symptomatic hypercalcaemia at four, respectively, five days postoperatively [15,16,17]. Late occurrence may be possible, as CCCM dissolve over weeks to months [21]. Dissolution of CCCM does not necessarily follow the release profile of added antibiotics, which may be much faster [5,18]. The calcium burden is high (1 mL of CaSO_4_ weights 1.5–3 g, depending on porosity, Ca^++^ representing approximately 1/4 of the weight) [21].

Other factors may precipitate decompensation (Figure 3). Patients with PJI or FRI often are elderly, restricted in their mobility and have medications for other health issues interfering with the calcium metabolism. This patient group is already at risk of developing hypercalcaemia. Implantation of a CCCM might then overload compensatory mechanisms and induce clinically relevant hypercalcaemia. The calcium metabolism is regulated by the gastrointestinal tract, the bone and the kidney, all being influenced by age. Especially, the buffering capacity of the bone decreases with age [22,23,24]. The main route of calcium elimination is renal excretion [25]. A reduction in calcium excretion may result from decreased glomerular filtration as well as from an increase in tubular reabsorption. The renal function decreases with age, as does the intestinal calcium resorption [26]. This somehow counteracts the lowered capacity of the bone to buffer calcium and of the kidney to excrete it. With increased calcium intake or vitamin D supplementation, the protective effect of reduced intestinal absorption may be lost [27]. Thiazides, angiotensin-converting enzyme inhibitors, nonsteroidal anti-inflammatory agents (NSAID), loop diuretics and low sodium diets all can reduce glomerular filtration, which in turn reduce the capacity to excrete calcium [27,28,29]. While the second case was an elderly and frail patient who can intrinsically be considered at risk, the first patient presented a peculiar risk factor: the habit of drinking three litres of milk every day. Hypercalcemia after excessive intake of cream or milk is known as the Burnett-Syndrome, or milk–alkali-syndrome [27]. This was reported in the past, following treatment of gastric ulcerations, before more potent antacids were available. In both patients, the therapy was to stop supplementary calcium intake, performing intravenous hyper-hydration for several days, and to block osteoclastic activity with intravenous bisphosphonates. Calcitonin induces rapid, but short-lasting deposition of calcium in bone [20].

Other severe complications are much more frequent in the treatment of orthopaedic infections. Antimicrobial-related severe adverse events affect 15% of patients treated for bone and joint infections caused by *S. aureus*, higher age and administration of rifampin being particular risk factors [30]. A severe daptomycin-induced adverse reaction was observed in Case Two. Severe adverse reactions to daptomycin are reported in more than 10% of treatments [31,32,33,34]. Other major complications such as instability/dislocation, intraoperative periprosthetic fracture, perioperative mortality, thromboembolism and infection recurrence have much higher rates after revision THA than the observed risk of hypercalcaemia [35,36,37,38,39]. Considering the risk of infection recurrence in the treatment of osteomyelitis, fracture-related infections (FRI) and PJI, the use of CaSO_4_ as carrier material for local application of antibiotics might then well be justified to enhance antimicrobial therapy. With an incidence of close to 1%, severe and symptomatic hypercalcemia should, however, be considered as a possible complication of CCCM applications.

After these two cases of severe symptomatic hypercalcemia, we began to systematically monitor calcium levels. The result of this monitoring will be published in a follow-up study as soon as possible. Additionally, the maximum quantity of CaSO_4_ implanted was reduced according to renal function. Previously, only the quantity of antibiotics was adapted, choosing the volume of CaSO_4_ solely based on the volume of dead space to obliterate [5,18]. When the preoperative renal function is severely impaired (eGFR < 30 mL/min), CaSO_4_ should be limited to 50 mL for Osteoset, equivalent to 20 mL of Stimulan (Biocomposites, Keele, UK). We further recommend stopping or modifying any medication interfering with the calcaemia, such as thiazide diuretics, treatment with vitamin D or calcium supplementation. NSAIDs may also be withheld. The monitoring of total calcium levels in the blood starts preoperatively and ends when the CCCM have radiologically resorbed. Protein complements should be used carefully, as they contain further calcium. Dietary habits are also to be verified.

Monitoring of calcaemia can be performed either by measuring ionized calcium (as usually available in blood gas analysis) or by measuring total serum calcium. Measurement of ionized calcium is only an approximation based on several assumptions and is affected by numerous variables such as pH, serum protein content and others [40]. This allows only a crude identification of hypo- and hypercalcaemia. Measurement of total serum calcium is much more accurate. The cut off for severe symptomatic hypercalcaemia can be made at 3.5 mmol/L [20]. As calcaemia is affected by serum protein content, especially albumin, correction of total calcium for albuminaemia is an easy and cost-effective solution [40].

Despite inherent risks, CaSO_4_ remains a very interesting antibiotic carrier material, providing a slow and long-lasting local release of antibiotics with relatively low systemic complication rates compared to systemic application [5,30]. Implantation of CCCM exposes the patient to large quantities of calcium, and this may induce hypercalcaemia during the dissolution and resorption process. Although the risk of associated severe hypercalcaemia appears to be rather low (<1%), potential risk factors should be optimized preoperatively. Dissolution kinetics and risk of hypercalcaemia may vary depending on the CCCM used, even among CaSO_4_. Considering the difficulties to identify clinical signs of hypercalcaemia and considering the potential severity of this complication, postoperative monitoring of serum calcium may be recommended.

## Figures and Tables

**Figure 1 jcm-11-04900-f001:**
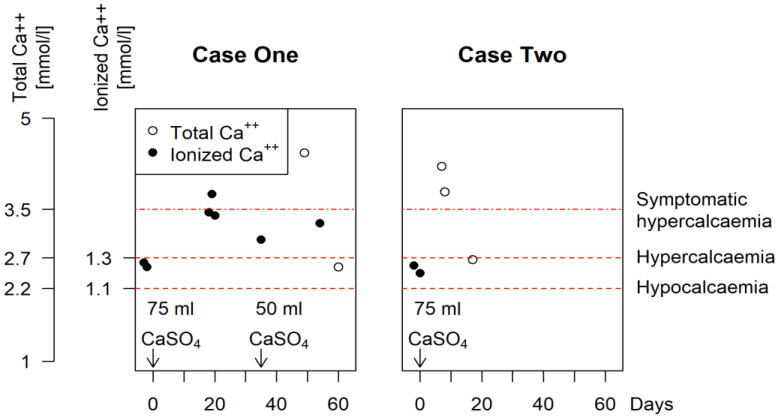
Perioperative calcium blood level measurements of the two cases affected by symptomatic hypercalcaemia. Black dots indicate the ionized calcium obtained with blood gas analysis. Considering the influence of pH on the levels of ionized calcium, this allows only a crude identification of hypo- and hypercalcemia. In Case One, we see ionized calcium levels high above the commonly admitted symptomatic level, but the patient presented no symptoms at that time. The white dots show the values of albumin-corrected total calcium levels. A line marks the usually accepted threshold for severe symptomatic hypercalcemia of 3.5 mmol/L. The CaSO_4_ dose is noted in ml, as defined by the manufacturer of the product used (Osteoset, Wright Medical, Memphis, TN, USA). In Case One, vancomycin-loaded calcium sulphate beads (CaSO_4_) had been implanted two times during a staged procedure. Case Two was a single-stage revision with debridement and exchange of the modular components.

**Figure 2 jcm-11-04900-f002:**
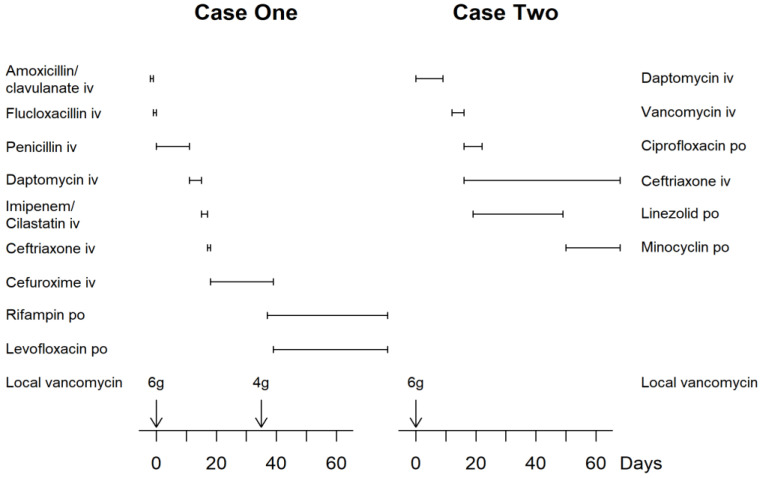
Summary of the systemic and local antibiotic therapy given over time in the two cases affected by symptomatic hypercalcaemia. Case One had developed interstitial nephritis secondary to administration of penicillin. Case Two had developed neurotoxicity as well as rhabdomyolysis secondary to administration of daptomycin. Both adverse reactions were entirely reversible.

**Figure 3 jcm-11-04900-f003:**
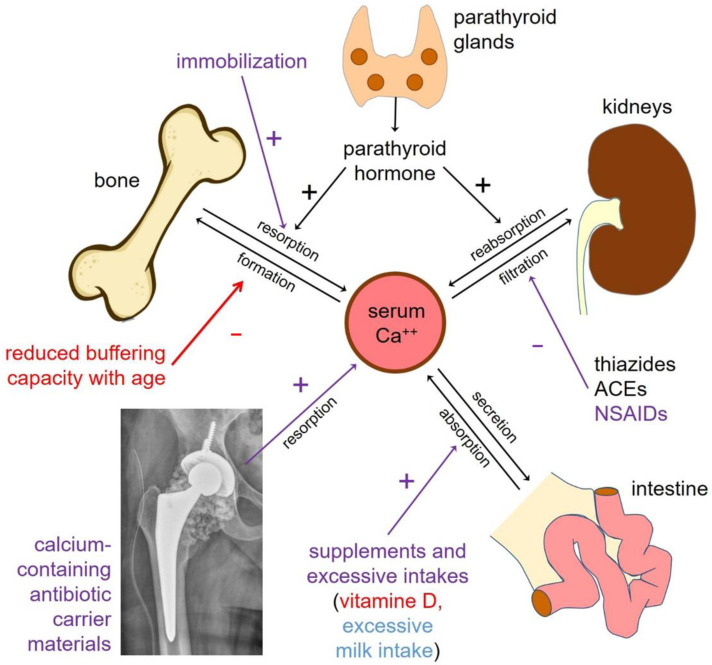
Overview of the pathophysiologic pathways involved in calcium homeostasis. Disturbing factors present in the two cases described are marked. Factors affecting only Case One are marked blue, those affecting only Case Two are marked red and those present in both cases are marked purple. Implanted CaSO_4_ represents a large calcium burden for the patient, as calcium makes out approximately ¼ of the weight of the material. Decreased compensatory mechanism in elderly or frail patients may then be overwhelmed, particularly if any drugs cause supplementary interference.

**Table 1 jcm-11-04900-t001:** Demographic and surgical data of the patient series collected.

Applications (n)	215	Case One	Case Two
**Gender, n (%)**	138 (64) male; 77 (36) female	male	female
**Age, median (range)**	69 years (5 to 90)	68 years	82 years
**Site of infection, n (%)**	Upper limb 11 (5)	Hip	Hip
	Spine 2 (1)		
	Pelvis 9 (4)		
	Hip 124 (58)		
	Thigh 15 (7)		
	Knee 15 (7)		
	Lower leg 28 (13)		
	Foot and ankle 11 (5)		
**Type of infection, n (%)**	PJI ^1^ 127 (59)	Native joint septic arthritis	PJI
	FRI ^2^ 82 (38)		
	Native joint septic arthritis 2 (1)		
	Others ^3^ 4 (2)		
**Renal function (eGFR), median (range)**	95 mL/min/1.7 m^2^ (9–272)	50	68
**Quantities of CaSO_4_, median (range)**	Ostoset 50 mL (5–250)	Osteoset 75 + 50 mL (staged procedure)	Osteoset 75 mL
**Quantities of CaSO_4_ grouped, n (%)**	≤25 mL: 62 (29)	>75 mL	≤75 mL
	>25 to ≤50 mL: 71 (33)		
	>50 to ≤75 mL: 71 (33)		
	>75 mL: 11 (5)		

^1^ PJI: periposthetic joint infection, ^2^ FRI: fracture related infections, ^3^ others: soft tissue, bone defects.

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
