# Peer review of "The Incidence of Severe Hypercalcaemia-Induced Mental Status Changes in Patients Treated with Antibiotic-Loaded Calcium Sulphate Depot for Orthopaedic Infections"

_jcm, 2022, doi:10.3390/jcm11164900_

Round 1
Reviewer 1 Report
Overall response
The authors of this manuscript have an exceptional topic to present: the incidence and management of severe hypercalcemia-induced mental status changes of patients treated with CaSO4 antibiotic depot for orthopedic-related infection. This is what their title should be, and this is what their Purpose/Aim statement should be. It should also be what they state throughout their manuscript. The authors have a very important topic that is nearly completely undocumented in literature. Such a manuscript will be very helpful in guiding both counseling and adverse events.
Unfortunately, the authors refused to incorporate my most important suggestions from the first submission. The authors remain reluctant to explicitly state the purpose of the patient registry which enrolled the patients who were eventually evaluated. See my detailed feedback below. The authors also refuse to clearly state to the readers that only patients who have devastating complications that require ICU admission were assessed, possibly because patients with minor and treatable symptoms would likely have presented and were managed by more local hospitals. This oversight CANNOT be overlooked, as it is CRITICAL to understanding the scope of the problem. The authors may only see the tip of the iceberg and refuse to acknowledge this in the manuscript.
There are other relatively small quibbles I have. But the issues with the Methods cannot be overlooked. This is not an easy topic to report on. And this may be the best that is available. Which is totally fine. But it is inappropriate to not be straight forward with data and methods. Report the truth as truthfully as possible. Do not hide the imperfections, as they allow appropriate contextualization of results.
Abstract
-it is rather indignifying to refer to a human patient as “an organism.” Beyond that, all the bacteria in the patient’s body could be considered “organisms” and we don’t seem to care much about their health, as we are killing them with antibiotics. So please use normal words when referring to human patients.
Introduction
-here, and for all time, please do not use the word “organism” to refer to human patients. This may be reasonable in other languages. In English, there is a word for human patients. It is “a patient” or “patients.” Use that term. “Organism” can refer to ANY living entity, and dehumanizes a patient. See above comments from Abstract.
The authors present the patient cohort in a way that deceives the reader. You write that this data is taken from a “prospective collected series with over 200 applications.” While this may be true, the prospective series was not focused on hypercalcemia after CCCM. And that would be the assumption of any reader. It was previously recommended to very clearly specify that what you actually did was retrospectively review a cohort of patients, which was comprised of a prospective assessment (perhaps a registry?) of (what was the prospective data collection focused on? Despite my instruction to identify this, you refused to do so)
You do have a clear “aims” statement: “It was the aim of the current retrospective study to identify cases of 46 symptomatic hypercalcaemia potentially related to application of CCCM for the treatment 47 of bone and joint infections in a prospectively collected series with over 200 applications.” However, subsequently in the article you rephrase the outcome to “severe” in the Discussion 176 “It would thus be reasonable to assume all *severe* symptomatic cases should have been identified.” So this is misleading. Are you looking to identify SEVERE hyperca which leads to ICU admission and severe mental deterioration? Or are you looking for symptomatic hypercalcemia such as constipation, anorexia, nausea, other gastrointestinal issues, polyuria, dehydration, cardiac dysrhythmia, and muscular/bone pain?
Methods
The authors have marginally improved the methods. However, it seems clear to me that they are trying to over-value their work rather than state the extent to which is is valid. There is no shame in doing the best you can do, especially for something that is rare, as this is. However, the authors continue to obfuscate despite my direct instructions to be forthright. You methods are one paragraphs ling, fewer than 150 words. My critique of your Methods is longer than your actual Methods statement. I instructed you to explicitly state THE INCLUSION CRITERIA AND MOTIVATION of the CCCM patient registry. Either there is difficulty with the English writing, or there is intent to hide the motivation of the registry. I would think the first sentences should be “This is a retrospective analysis of a prospectively maintained patient registry. The registry enrolls all patients in whom CaSO4 (Osteoset…) had been implanted as carrier material for local delivery of antibiotics in the treatment of various orthopaedic infections. The dates of inclusion were October 2006 to February 2019.” I understand this may not be interpreted by the authors as different from what they wrote, but for a native English reader there is improved clarity. You must also state whether any patients in the registry were excluded, and for what reasons.
The next important point of the Methods must clearly, explicitly state how patients were identified as symptomatic. “The definition of symptomatic hypercalcemia for this study is … (for example, exactly as you wrote in the Response but refused to insert in the manuscript “any patient suffering loss of consciousness, stuporous state or any other major alteration of consciousness.” This is exceptionally important, because this is how you determined your patients. You did not review charts for stones groans bones, but only psychiatric overtones (to invoke a common English mnemonic). All those are clinical symptoms, but you did not evaluate any other than the altered mental status. I cannot emphasize enough *do NOT use the word “vigilance”!*
Following the identification of patients with mental status changes, further chart review then was performed to ensure the causative etiology was hypercalcaemia. Postoperative clouding of consciousness is caused mainly by opiates. This should however not have caused missing relevant cases, as recovery from opiate overhang is relatively short, whereas hypercalcaemia following application of CCCM occurs late.
Case One
See earlier comments regarding referring to patients as organisms.
Discussion
Because your Methods are misleading, your statement “symptomatic hyper-162 calcaemia was observed in only 2 out of 215 (0.9%) applications…” is also potentially grossly incorrect. You need to be exceptionally more forthright with your Methods. You state it in your Reviewer Response but not your manuscript. You MUST state “our hospital is a regional referral hospital. Patients normally present with issues such as post-operative adverse events to their regional hospital, and are referred to our hospital only when those hospitals must elevate care.” This is EXCEPTIONALLY important in this series because your only two patients who were identified were so devastatingly impacted that they required ICU care. I wrote this in the first response but you ignore this importance. Patients who had mild hypercalcemic symptoms very well may have been treated by the less intense hospitals, and you have no idea because you do not have access to their records. If you DO have access to their records, you need to report whether you looked at those records, and why you feel confident none met threshold for this inclusion in this study.
Stating “Extending the study period would 236 reduce the incidence” 236 is a bizarre statement to make. Do you want to over-report the incidence? You should report the data you have, and conclude with that.
Author Response
Reviewer 1
The authors of this manuscript have an exceptional topic to present: the incidence and management of severe hypercalcemia-induced mental status changes of patients treated with CaSO4 antibiotic depot for orthopedic-related infection. This is what their title should be, and this is what their Purpose/Aim statement should be. It should also be what they state throughout their manuscript. The authors have a very important topic that is nearly completely undocumented in literature. Such a manuscript will be very helpful in guiding both counseling and adverse events.
Thank you very much for this positive appraisal of our work. We gladly take over the proposed title. We would only propose a slight modification, to avoid abbreviations in the title and to shorten it a bit more:
"The incidence and management of severe hypercalcaemia-induced mental status changes in patients treated with antibiotic-loaded calcium sulphate depot for orthopaedic infections"
Unfortunately, the authors refused to incorporate my most important suggestions from the first submission.
Please accept our apologies for the time and effort it took you to review our manuscript, as well as for the process of resubmission. We resubmitted as we simply did not figure out what should be modified explicitly. The details you provide now are very helpful in better understanding how this manuscript may be improved. Sorry for having missed to better formulate in the manuscript some points of discussion mentioned in the reply to the reviewers.
The authors remain reluctant to explicitly state the purpose of the patient registry which enrolled the patients who were eventually evaluated. See my detailed feedback below. The authors also refuse to clearly state to the readers that only patients who have devastating complications that require ICU admission were assessed, possibly because patients with minor and treatable symptoms would likely have presented and were managed by more local hospitals. This oversight CANNOT be overlooked, as it is CRITICAL to understanding the scope of the problem. The authors may only see the tip of the iceberg and refuse to acknowledge this in the manuscript.
There are other relatively small quibbles I have. But the issues with the Methods cannot be overlooked. This is not an easy topic to report on. And this may be the best that is available. Which is totally fine. But it is inappropriate to not be straight forward with data and methods. Report the truth as truthfully as possible. Do not hide the imperfections, as they allow appropriate contextualization of results.
We highly appreciate the clarifications provided. We also fully acknowledge the importance of proper description of methods. We hope we could provide adequate precision in the Patients and Methods section. Please refer to the extensive modifications of the Patients and Methods section.
Abstract
it is rather indignifying to refer to a human patient as “an organism.” Beyond that, all the bacteria in the patient’s body could be considered “organisms” and we don’t seem to care much about their health, as we are killing them with antibiotics. So please use normal words when referring to human patients.
The word "organism" has been replaced by "patient" (page 1 line 19). We certainly did not intend to be indignifying.
Introduction
here, and for all time, please do not use the word “organism” to refer to human patients. This may be reasonable in other languages. In English, there is a word for human patients. It is “a patient” or “patients.” Use that term. “Organism” can refer to ANY living entity, and dehumanizes a patient. See above comments from Abstract.
The same correction was provided throughout the manuscript (page 1 line 41, page 7 line 228 in the legend of Fig. 3, page 8 line 272).
The authors present the patient cohort in a way that deceives the reader. You write that this data is taken from a “prospective collected series with over 200 applications.” While this may be true, the prospective series was not focused on hypercalcemia after CCCM. And that would be the assumption of any reader. It was previously recommended to very clearly specify that what you actually did was retrospectively review a cohort of patients, which was comprised of a prospective assessment (perhaps a registry?) of (what was the prospective data collection focused on? Despite my instruction to identify this, you refused to do so)
Sorry for the misunderstanding. In Patients and Methods, it already was stated that a retrospective study was performed on a prospectively collected cohort of patients. Further clarifications were provided regarding the exact screening of patient files performed. Please refer to the section Patients and Methods for the modifications made. We sincerely hope to have provide adequate precision.
You do have a clear “aims” statement: “It was the aim of the current retrospective study to identify cases of symptomatic hypercalcaemia potentially related to application of CCCM for the treatment of bone and joint infections in a prospectively collected series with over 200 applications.” However, subsequently in the article you rephrase the outcome to “severe” in the Discussion “It would thus be reasonable to assume all *severe* symptomatic cases should have been identified.” So this is misleading. Are you looking to identify SEVERE hyperca which leads to ICU admission and severe mental deterioration? Or are you looking for symptomatic hypercalcemia such as constipation, anorexia, nausea, other gastrointestinal issues, polyuria, dehydration, cardiac dysrhythmia, and muscular/bone pain?
Thank you for having pointed out potentially misleading elements we had overseen. The primary aim of the study was to identify severe/symptomatic hypercalcaemia. We identified two cases of mental status alteration and no case of hypercalcaemia-associated dysrhythmias. This was clarified in the Introduction (page 2 line 49-50) as well as in the Discussion (page 5 line 170-1). That less severe or less specific symptoms are impossible to identify retrospectively in such a patient cohort has been clarified in the second paragraph of the Discussion (page 6 line 177-8). We sincerely hope we could provide the necessary clarifications. The methods have limitations, but we have nothing to hide and sincerely believe the cases described are instructive.
Methods
The authors have marginally improved the methods. However, it seems clear to me that they are trying to over-value their work rather than state the extent to which is is valid. There is no shame in doing the best you can do, especially for something that is rare, as this is. However, the authors continue to obfuscate despite my direct instructions to be forthright. You methods are one paragraphs ling, fewer than 150 words. My critique of your Methods is longer than your actual Methods statement. I instructed you to explicitly state THE INCLUSION CRITERIA AND MOTIVATION of the CCCM patient registry. Either there is difficulty with the English writing, or there is intent to hide the motivation of the registry. I would think the first sentences should be “This is a retrospective analysis of a prospectively maintained patient registry. The registry enrolls all patients in whom CaSO4 (Osteoset…) had been implanted as carrier material for local delivery of antibiotics in the treatment of various orthopaedic infections. The dates of inclusion were October 2006 to February 2019.” I understand this may not be interpreted by the authors as different from what they wrote, but for a native English reader there is improved clarity. You must also state whether any patients in the registry were excluded, and for what reasons.
We gladly take over the proposed text regarding description of the study/registry. As mentioned, we did not perceive the importance of the formulation, but most happily take over the suggestion. Please refer to the beginning of Patients and Methods. All patients included had provided written informed consent regarding data analysis, an essential aspect when implanting antibiotic-loaded CCCM, at least in the early phase of the study, when this method was not as well established as it is nowaydays.
The next important point of the Methods must clearly, explicitly state how patients were identified as symptomatic. “The definition of symptomatic hypercalcemia for this study is … (for example, exactly as you wrote in the Response but refused to insert in the manuscript “any patient suffering loss of consciousness, stuporous state or any other major alteration of consciousness.” This is exceptionally important, because this is how you determined your patients. You did not review charts for stones groans bones, but only psychiatric overtones (to invoke a common English mnemonic). All those are clinical symptoms, but you did not evaluate any other than the altered mental status. I cannot emphasize enough *do NOT use the word “vigilance”!*
There again, we most gladly take over the suggestion. We had tried to adapt the suggestions as good as possible, considering other general requirements, such as overall length of the manuscript. Please refer to the rewritten section Patients and Methods as well as the first paragraph of the Discussion, page 5, for the text modifications.
Following the identification of patients with mental status changes, further chart review then was performed to ensure the causative etiology was hypercalcaemia. Postoperative clouding of consciousness is caused mainly by opiates. This should however not have caused missing relevant cases, as recovery from opiate overhang is relatively short, whereas hypercalcaemia following application of CCCM occurs late.
We totally agree. We expanded Patients and Methods, as well as the second paragraph of the Discussion, to better point this out. The most limiting factors is the retrospective analysis, as discussed above and hopefully better pointed out in the modified manuscript.
Case One
See earlier comments regarding referring to patients as organisms.
As for the previous mentions, "organism" was replaced by "patient", following the reviewers suggestion.
Discussion
Because your Methods are misleading, your statement “symptomatic hyper-calcaemia was observed in only 2 out of 215 (0.9%) applications…” is also potentially grossly incorrect. You need to be exceptionally more forthright with your Methods. You state it in your Reviewer Response but not your manuscript. You MUST state “our hospital is a regional referral hospital. Patients normally present with issues such as post-operative adverse events to their regional hospital, and are referred to our hospital only when those hospitals must elevate care.” This is EXCEPTIONALLY important in this series because your only two patients who were identified were so devastatingly impacted that they required ICU care. I wrote this in the first response but you ignore this importance. Patients who had mild hypercalcemic symptoms very well may have been treated by the less intense hospitals, and you have no idea because you do not have access to their records. If you DO have access to their records, you need to report whether you looked at those records, and why you feel confident none met threshold for this inclusion in this study.
We hope we could provide this time the modifications and explanations necessary. Please refer to the comments above and the modifications mentioned in the Patients and Methods and Discussion sections. We totally agree with the reviewer regarding the point of criticism, but we also believe any reader should be able to figure this out from the information provided. Nevertheless, we believe having provided honest and truthful data. One particularity that might be of interest to the reviewer, but which can't be reasonably written in a scientific publication, is, that in Switzerland regional hospitals do not commonly treat postoperative complications, but transfer the patients to the center the operation was performed in. Distances are short in our country. In case of lack of capacity, patients may stay in the regional hospitals. But hospitalization reports are usually provided and would have been available in the patient files.
Stating “Extending the study period would reduce the incidence” is a bizarre statement to make. Do you want to over-report the incidence? You should report the data you have, and conclude with that.
We propose to simply remove the problematic sentence. All we wanted to point out was that the incidence being low, it is influenced by the observation period.
Reviewer 2 Report
The authors present an analysis of their prospective series of patients treated with calcium containing carrier materials, specifically a formulation of CaSO4, with respect to the occurrence of symptomatic hypercalcemia. In this series, which is summarized in general, they note an incidence of this complication of 0.9%, with other factors noted to contribute to the condition, which is acknowledged to not solely arise from the use of CaSo4. The 2 patients noted to make up the 0.9 % incidence are presented in the format of abbreviated case reports, allowing the authors to fully explore the circumstances of the condition. This is a notable and novel topic as only individual case reports have previously noted the complication and prior work has not considered its incidence in a larger series (215 patients total), nor delved into the many possible contributing risk factors and co-morbidities.
The methods are appropriate to a specific analysis of an existing patient series (retrospective chart review) based on symptoms. The author acknowledged that their monitoring was strictly clinically driven, and did not take into account possible asymptomatic cases which might have had similar laboratory findings if evaluated.
The presentation of the individual symptomatic cases adds context, interest and relevance to the notation of a fairly uncommon complication. The case reports would be easier to follow if a little more organized with more specific details of the involved timeline and events. (lines 90-127, 128-153).
I do have a question about the ethical background of the article. I see no mention of consent or the need for consent. What patient protections/permissions are in place to allow the use of data both for the specific case reports and for the prospective patient series from which they were taken? I would like to see this mentioned and many journals and institutions require this.
Both figure 1 and figure 2 present a large amount of data. This is initially somewhat difficult to follow. Would the authors be able to reconstruct the figures, possibly adding another figure to simplify the existing figures and better demonstrate their findings?
In general the discussion is relevant and helpful to understand the conditions discussed, which are important to consider when using the Calcium containing carrier materials. Due to the lack of background testing of Calcium levels in the absence of symptoms, there is necessarily speculation regarding the significance of the laboratory findings as related to the use of the calcium sulfate, and even the attribution of the symptoms to the hypercalcemic state. There is also no summary of incidence of the risk factors or potentially contributory co- morbidities in the greater series-- Can this be included in the article? I would suggest a follow up study employing more intentional calcemic monitoring and accounting for risk factors, to define clinical significance and reduce speculation. (However, I would not require this as part of current publication.)
In summary, the authors bring up an important concept based on their data, and discuss it thoroughly. I believe this paper fills a gap in the general knowledge and practice among those using Calcium containing carrier materials, and serves as a cautionary tale, educating the readers to risk factors and co-contributing conditions, and to have a low threshold for further monitoring of calcium levels. Future studies could prospectively evaluate contributing factors in further details and involve control patients to comparatively monitor for adverse events and refine the attribution of the symptoms to their causes.
Author Response
Reviewer 2
The authors present an analysis of their prospective series of patients treated with calcium containing carrier materials, specifically a formulation of CaSO4, with respect to the occurrence of symptomatic hypercalcemia. In this series, which is summarized in general, they note an incidence of this complication of 0.9%, with other factors noted to contribute to the condition, which is acknowledged to not solely arise from the use of CaSo4. The 2 patients noted to make up the 0.9 % incidence are presented in the format of abbreviated case reports, allowing the authors to fully explore the circumstances of the condition. This is a notable and novel topic as only individual case reports have previously noted the complication and prior work has not considered its incidence in a larger series (215 patients total), nor delved into the many possible contributing risk factors and co-morbidities.
Thank you very much for this appraisal of our manuscript. That was exactly the motivation we had to try to publish these findings.
The methods are appropriate to a specific analysis of an existing patient series (retrospective chart review) based on symptoms. The author acknowledged that their monitoring was strictly clinically driven, and did not take into account possible asymptomatic cases which might have had similar laboratory findings if evaluated.
The methods indeed have some limitations. Please refer to the comments of Reviewer 1. There have been some additions to the section Patients and Methods, following his suggestions.
The presentation of the individual symptomatic cases adds context, interest and relevance to the notation of a fairly uncommon complication. The case reports would be easier to follow if a little more organized with more specific details of the involved timeline and events. (lines 90-127, 128-153).
We acknowledge the point raised by the reviewer. That was the motivation for presentation detailed information in Table 1, as well as presenting the course of calcium serum levels in Fig. 1 and the antibiotics administered in Fig. 2. The operations are marked in both figures to ease interpretation. We had to keep the text short to fulfil the requirements for maximum word count of the manuscript. Please accept our apologies that we didn't figure out how to improve presentation further within the time frame of 5 days given for revision.
I do have a question about the ethical background of the article. I see no mention of consent or the need for consent. What patient protections/permissions are in place to allow the use of data both for the specific case reports and for the prospective patient series from which they were taken? I would like to see this mentioned and many journals and institutions require this.
Sorry for having missed mentioning this. Signed informed consent was available from both patients and obtained before implantation of antibiotic-loaded CaSO4. This information was added in Patients and Methods
Both figure 1 and figure 2 present a large amount of data. This is initially somewhat difficult to follow. Would the authors be able to reconstruct the figures, possibly adding another figure to simplify the existing figures and better demonstrate their findings?
We acknowledge this point, too. It wasn't easy to establish appropriate figures, particularly as quite some parameters have to be considered in such cases. The figures are a bit crowded indeed. We chose to provide information from both cases in similar figures provided together, to ease comparison. However, we are sorry not having figured out a better way of presentation within the delay of 5 days provided for minor revision. In case this remains required, we would be open to suggestions how to improve presentation.
In general, the discussion is relevant and helpful to understand the conditions discussed, which are important to consider when using the calcium containing carrier materials. Due to the lack of background testing of calcium levels in the absence of symptoms, there is necessarily speculation regarding the significance of the laboratory findings as related to the use of the calcium sulfate, and even the attribution of the symptoms to the hypercalcemic state. There is also no summary of incidence of the risk factors or potentially contributory co-morbidities in the greater series-- Can this be included in the article? I would suggest a follow up study employing more intentional calcemic monitoring and accounting for risk factors, to define clinical significance and reduce speculation. (However, I would not require this as part of current publication.)
We fully agree with the reviewer regarding this point. The findings of this study prompted us to start systematic monitoring with prospective data collection, as it is impossible to reconstruct retrospectively all the data necessary for a thorough analysis of risk factors. Particularly as patient inclusion started in 2006, and as electronic patient files where not yet available. May a promise to provide the required study as soon as possible be enough to answer this?
In summary, the authors bring up an important concept based on their data, and discuss it thoroughly. I believe this paper fills a gap in the general knowledge and practice among those using calcium containing carrier materials, and serves as a cautionary tale, educating the readers to risk factors and co-contributing conditions, and to have a low threshold for further monitoring of calcium levels. Future studies could prospectively evaluate contributing factors in further details and involve control patients to comparatively monitor for adverse events and refine the attribution of the symptoms to their causes.
Thank you very much for this comment. We strongly believe this. We just couldn't say it ourselves. Thank you for having taken the time to review our manuscript.
"Please see the attachment"

This manuscript is a resubmission of an earlier submission. The following is a list of the peer review reports and author responses from that submission.
Round 1
Reviewer 1 Report
I think it is a well-written retrospective analysis aiming at establishing the incidence of symptomatic hypercalcemia antibiotic-loaded calcium sulphate.
The possible shortcoming arises from the fact that only 2 cases were identified among 215 cases and we cannot be sure about the true incidence of this complication. The authors address this limitation in the Discussion and in the title referring to their study as a preliminary report. All possible contributing factors to hypercalcemia in two cases are explained in detail. Also the graphics is self-explanatory,
Author Response
We would like to thank the reviewer for this very positive evaluation. It was our desire to provide the reader with such a self-explanatory publication, despite the limited evidence available.
Reviewer 2 Report
The article describes a series of 215 patients who were evaluated to assess the frequency of symptomatic hypercalcemia following CCCM depot. The authors report 2 events, for a rate of .9%. The topic is interesting and potentially very clinically relevant, but the main learning point of the paper as currently written is merely that “ICU admission for hypercalcemia crisis following CCCM is rare.” This is well known, and the current paper as written does not seem to impact knowledge or bring a forgotten point to the fore. Further, I have concerns regarding the methodology. Specifically. 1) The authors are not clear about the source of patients evaluated: it is a registry, but I could not clearly identify the inclusion criteria of the registry (was it ALL patients who had CCCM?). 2) The authors are not clear about what signs symptoms etc were looked at during the retrospective chart review; the only two patients they state as clinically symptomatic were admitted to the ICU, which is far beyond a clinically apparent threshold of hypercalcemia and would be better described as having a hypercalcemic crisis. This makes the screening criteria seem to be “clinically relevant hypercalcemia was defined as deterioration requiring ICU workup.” 3) The authors modified their routines to include a monitoring protocol, which is to be commended; this appears to have eliminated ICU admissions, probably achieving its goal. This current manuscript, which appears to be a case report of two patients with an unclear reason to report a denominator of 215 patients other than to improve the immediate appearance, seems to be a preamble for the more useful paper they intend to write. I recommend instead that the current manuscript is deferred in favor of the upcoming paper which is likely much more clinically helpful. That paper could be titled “implementation of a laboratory study calcium monitoring protocol eliminates hypercalcemic crisis: a comparison of 2 cohorts of 215 patients.” If that paper identifies physical exam findings that were specifically screened in the notes (even if they were only specifically commented on after the monitoring phase), the calcium levels meeting threshold in the monitoring phase, and the clinically subtle but identifiable warning signs identified during the monitoring phase, it would be tremendously helpful to surgeons and medical doctors caring for patients following CCCM.
—-
I am unclear regarding the stated methods. Symptomatic hypercalcemia after CCM is very rare, as highlighted by the authors in lines 41-46. Yet, the authors state they were alert enough to this potential issue to have started collecting data prospectively on over 200 patients, as early as October 2006. This would represent incredible foresight by the authors to have started a registry on this topic at that time. It would be more reasonable if these patients were part of a separate prospective registry and that registry was retrospectively evaluated, but this is not the statement from the authors. The prospective/retrospective nature of the patients series and data collection if further confused by the authors stating in line 46 “the aim of the current *retrospective* study...”. Please clarify the patient enrollment or selection process. Line 53-54 seems to hint at the registry enrollment process (perhaps all patients getting CCCM were enrolled?) but again, it is not clear enough. Please clarify what the registry was originally created for, how many patients were part of the registry, how the selection process was of the total patients was filtered to 215.
The authors state that a retrospective chart review was done to identify episodes of clinically symptomatic hypercalcemia. While not as ideal as routinely obtained laboratory values, this methodology is reasonable given the retrospective nature of the study and the low frequency of issues. However, the authors do not clearly identify what phrasing or words or features were specifically sought as indicating clinically relevant hypercalcemia. If it was based on the classic mnemonic “groans bones stones moans thrones psychiatric overtones” that is reasonable, but it must be stated in a clear and medically appropriate way. I would guess the chart is unlikely to say “clinical hypercalcemia” so there must have been some words, exam features, or history/update status words and phrases that cued the researchers to suspect or confirm clinically relevant hypercalcemia. Did zero other patients in the reviewed cohort express any of the potentially positive symptoms? Almost all patients must have had groans (being post-operative), so perhaps a combination of “groans” along with one or two other findings would be a reasonable way to establish a clinically relevant suspicion of hypercalcemia? Otherwise, the study appears to be at risk of being a “we remember two patients who had really terrible hypercalcemia requiring prolonged dramatic care, let’s write a paper of those two case reports and make it look better by having the denominator of patients be everyone in the database.” I am not trying to accuse the authors of doing this, but it is important to more clearly identify, justify, and verify the screening criteria to identify patients. Were there zero other patients who had relatively minor clinically apparent hypercalcemia?
Line 112, what does “vigilance reduction” mean? Perhaps it is a regional term, I am not familiar with that term.
Lines 230-245 are extremely important and helpful for this study. I commend the authors for recognizing an issue and implementing a system to prevent or minimize problems. However, the inclusion of the monitoring data in this study would tremendously improve the quality and clinical value of this current study. I believe the current study would be rather strong if the authors did a comparison study: cohort 1 is CCCM and clinically detected hypercalcemia BEFORE routine monitoring, and cohort 2 is CCCM and clinically detected hypercalemia AFTER routine monitoring. Given the protocol change, it is likely the clinicians were more directly evaluating patients for clinical symptoms of hypercalcemia, which although limiting the comparability of the cohorts, would be acceptable given the rarity of the problem. Again, the specific clinical features to identify hypercalcemia is critical. Currently, the only clear criteria it seems is “admission to the ICU with subsequent expansive workup identifying hypercalcemia.” It seems reasonable that no patients were “clinically hypercalcemic” after the implementation of the monitoring protocol because the hypercalcemia was detected before patients deteriorated to ICU requirement.
Author Response
Reviewer 2
The article describes a series of 215 patients who were evaluated to assess the frequency of symptomatic hypercalcemia following CCCM depot. The authors report 2 events, for a rate of .9%. The topic is interesting and potentially very clinically relevant, but the main learning point of the paper as currently written is merely that “ICU admission for hypercalcemia crisis following CCCM is rare.” This is well known, and the current paper as written does not seem to impact knowledge or bring a forgotten point to the fore.
We fully acknowledge the limitations of the methods. In our experience, the issue of hypercalcaemia following application of CCCM is not well known among users. Isolated case reports have been published about this suibject, however the issue is not known generally. We also made our experiences over the years, becoming more careful after having encountered certain complications. Nevertheless, as we gathered experience over quite an extensive series of cases, we thought it useful to provide the community with preliminary data to better assess the risks of hypercalcaemia after application of CCCM. Being a complication affecting not even 1% of the cases, it remains rare. Particularly compared to other complications in treatment of orthopaedic device-associated infections. Affecting 1% of the cases, symptomatic hypercalcaemia should however be considered as one of the usual complications of CCCM, going beyond isolated case reports, and patients should be informed appropriately. We believe this information is important for clinicians. The manuscript has been modified accordingly to underscore the clinical relevance.
Further, I have concerns regarding the methodology. Specifically. 1) The authors are not clear about the source of patients evaluated: it is a registry, but I could not clearly identify the inclusion criteria of the registry (was it ALL patients who had CCCM?). 2) The authors are not clear about what signs symptoms etc were looked at during the retrospective chart review; the only two patients they state as clinically symptomatic were admitted to the ICU, which is far beyond a clinically apparent threshold of hypercalcemia and would be better described as having a hypercalcemic crisis. This makes the screening criteria seem to be “clinically relevant hypercalcemia was defined as deterioration requiring ICU workup.” 3) The authors modified their routines to include a monitoring protocol, which is to be commended; this appears to have eliminated ICU admissions, probably achieving its goal. This current manuscript, which appears to be a case report of two patients with an unclear reason to report a denominator of 215 patients other than to improve the immediate appearance, seems to be a preamble for the more useful paper they intend to write. I recommend instead that the current manuscript is deferred in favor of the upcoming paper which is likely much more clinically helpful. That paper could be titled “implementation of a laboratory study calcium monitoring protocol eliminates hypercalcemic crisis: a comparison of 2 cohorts of 215 patients.” If that paper identifies physical exam findings that were specifically screened in the notes (even if they were only specifically commented on after the monitoring phase), the calcium levels meeting threshold in the monitoring phase, and the clinically subtle but identifiable warning signs identified during the monitoring phase, it would be tremendously helpful to surgeons and medical doctors caring for patients following CCCM.
All patients treated in our institutions over the study period with antibiotic-loaded CCCM were included in a prospectively collected database. This permitted a reliable identification of the denominator. We agree with the reviewer that identification of the cases may not be based on hard criteria. Nevertheless, problematic hypercalcaemia should be identifiable reliably, even retrospectively. Those cases do not necessarily require admission to an ICU. Serving as regional reference hospitals, we would expect to have identified cases treated initially in peripheral hospital, as these would/should have been transferred to our facility in case once symptomatic. It might as well be argued that asymptomatic hypercalcaemia may not be relevant. We totally agree with the reviewer that a systematic analysis is necessary. Nevertheles, the aetiology being multifactorial, as presented in our manuscript, in a complex patient population, we believe our manuscript may be useful to clinicians taking care of orthopaedic device-associated infections. The manuscript has been modified to illustrate inclusion criteria more precisely.
I am unclear regarding the stated methods. Symptomatic hypercalcemia after CCM is very rare, as highlighted by the authors in lines 41-46. Yet, the authors state they were alert enough to this potential issue to have started collecting data prospectively on over 200 patients, as early as October 2006. This would represent incredible foresight by the authors to have started a registry on this topic at that time. It would be more reasonable if these patients were part of a separate prospective registry and that registry was retrospectively evaluated, but this is not the statement from the authors. The prospective/retrospective nature of the patients series and data collection if further confused by the authors stating in line 46 “the aim of the current *retrospective* study...”. Please clarify the patient enrollment or selection process. Line 53-54 seems to hint at the registry enrollment process (perhaps all patients getting CCCM were enrolled?) but again, it is not clear enough. Please clarify what the registry was originally created for, how many patients were part of the registry, how the selection process was of the total patients was filtered to 215.
This was a retrospective chart review of a prospectively collected case series, as already mentioned. The manuscript has been modified to better illustrate the methods used. Considering the novelty of using antibiotic-loaded CCCM to treat orthopaedic device-associated infections and potential benefits and risks, all cases were collected prospectively from 2006 on to be able to make any retrospective analyse required, depending on the outcomes of interest. Enrolment may thus be considered as complete, as made by the surgeons involved in treating these cases. Retrospective review of any specific outcome of course suffers limitations from possible identification in chart review. Symptomatic hypercalcaemia leads to disturbed vigilance, and this should be identifiable quite reliably in medical reports. We agree that theoretically the methods are weak. Nevertheless, it may be quite reliable, particularly as the hospitals involved serve as reference centre for ICU care.
The authors state that a retrospective chart review was done to identify episodes of clinically symptomatic hypercalcemia. While not as ideal as routinely obtained laboratory values, this methodology is reasonable given the retrospective nature of the study and the low frequency of issues. However, the authors do not clearly identify what phrasing or words or features were specifically sought as indicating clinically relevant hypercalcemia. If it was based on the classic mnemonic “groans bones stones moans thrones psychiatric overtones” that is reasonable, but it must be stated in a clear and medically appropriate way. I would guess the chart is unlikely to say “clinical hypercalcemia” so there must have been some words, exam features, or history/update status words and phrases that cued the researchers to suspect or confirm clinically relevant hypercalcemia. Did zero other patients in the reviewed cohort express any of the potentially positive symptoms? Almost all patients must have had groans (being post-operative), so perhaps a combination of “groans” along with one or two other findings would be a reasonable way to establish a clinically relevant suspicion of hypercalcemia? Otherwise, the study appears to be at risk of being a “we remember two patients who had really terrible hypercalcemia requiring prolonged dramatic care, let’s write a paper of those two case reports and make it look better by having the denominator of patients be everyone in the database.” I am not trying to accuse the authors of doing this, but it is important to more clearly identify, justify, and verify the screening criteria to identify patients. Were there zero other patients who had relatively minor clinically apparent hypercalcemia?
The chosen method may well be criticized the way the reviewer did it. We fully acknowledge these limitations. Symptomatic severe hypercalcaemia however is a dramatic condition, not a condition with slight symptoms missed during patient care and in documentation. Even if it had not been recognized at the time of patient care, admission to ICU, investigations or other diagnoses related to major alterations of consciousness would have allowed identifying cases. Identification was not limited to the diagnosis of symptomatic hypercalcaemia to be able to identify potential other cases. Of course, asymptomatic hypercalcaemia will have been missed. For this reason, we specified symptomatic severe hypercalcaemia throughout the manuscript as being the outcome parameter. A sentence has been added to Methods to clarify case identification.
Line 112, what does “vigilance reduction” mean? Perhaps it is a regional term, I am not familiar with that term.
The term has been chosen on purpose to be vague. The aim of the study was to identify any patient suffering loss of consciousness, stuporous state or any other major alteration of consciousness. Detailed chart review then permitting to identify causative hypercalcaemia. Postoperative clouding of consciousness is caused mainly by opiates. This should however not have caused missing relevant cases, as recovery from opiate overhang is relatively short, whereas hypercalcaemia following application of CCCM occurs late. To clarify terminology, "vigilance reduction" was exchanged for "altered vigilance".
Lines 230-245 are extremely important and helpful for this study. I commend the authors for recognizing an issue and implementing a system to prevent or minimize problems. However, the inclusion of the monitoring data in this study would tremendously improve the quality and clinical value of this current study. I believe the current study would be rather strong if the authors did a comparison study: cohort 1 is CCCM and clinically detected hypercalcemia BEFORE routine monitoring, and cohort 2 is CCCM and clinically detected hypercalemia AFTER routine monitoring. Given the protocol change, it is likely the clinicians were more directly evaluating patients for clinical symptoms of hypercalcemia, which although limiting the comparability of the cohorts, would be acceptable given the rarity of the problem. Again, the specific clinical features to identify hypercalcemia is critical. Currently, the only clear criteria it seems is “admission to the ICU with subsequent expansive workup identifying hypercalcemia.” It seems reasonable that no patients were “clinically hypercalcemic” after the implementation of the monitoring protocol because the hypercalcemia was detected before patients deteriorated to ICU requirement.
Symptomatic hypercalcaemia does not necessarily require ICU admission. Missed hypercalcaemia would however lead to severe disturbances of consciousness. This would lead to ICU admission, and would have allowed us to identify cases missed in peripheral hospitals, as our hospitals serve as reference centres for intensive care. These elements have been specified, as highly relevant for case identification, as the reviewer underscored.
Affecting around 1% of the cases, symptomatic hypercalcaemia may be considered as being rare. Particularly compared to other complications encountered during treatment of orthopaedic device-associated infections, as discussed. On the other hand, a complication affecting 1% of the patients is not that rare and should be considered in any application of CCCM. In our experience, awareness of the issue among orthopaedic surgeons is not that good. This represents in our opinion a very good argument to raise awareness with such a publication. Further refinement is required, but this already has been discussed in the manuscript. Up to now, this study would be the best evidence available regarding the incidence of hypercalcaemia after application of CCCM.